# Improving the Electrochemical Performance and Stability of Polypyrrole by Polymerizing Ionic Liquids

**DOI:** 10.3390/polym12010136

**Published:** 2020-01-06

**Authors:** Arko Kesküla, Ivo Heinmaa, Tarmo Tamm, Nihan Aydemir, Jadranka Travas-Sejdic, Anna-Liisa Peikolainen, Rudolf Kiefer

**Affiliations:** 1Intelligent Materials and Systems Lab, Institute of Technology, University of Tartu, Nooruse 1, 50411 Tartu, Estonia; arkokeskyla@gmail.com (A.K.); tarmo.tamm@ut.ee (T.T.); anna.liisa.peikolainen@ut.ee (A.-L.P.); 2National Institute of Chemical Physics and Biophysics, Akadeemia tee 23, 12618 Tallinn, Estonia; ivo@kbfi.ee; 3Polymer Electronics Research Center, School of Chemical Sciences, University of Auckland, Private Bag 92019, Auckland 1142, New Zealand; nayd972@aucklanduni.ac.nz (N.A.); j.travas-sejdic@auckland.ac.nz (J.T.-S.); 4MacDiarmid Institute for Advanced Materials and Nanotechnology, Wellington 6011, New Zealand; 5Faculty of Applied Sciences, Ton Duc Thang University, Ho Chi Minh City 700000, Vietnam

**Keywords:** PPyPIL, electropolymerization, ssNMR, blend structure, electrochemical performance

## Abstract

Polypyrrole (PPy) based electroactive materials are important building blocks for the development of flexible electronics, bio-sensors and actuator devices. As the properties and behavior of PPy depends strongly on the operating environment—electrolyte, solvent, etc., it is desirable to plant immobile ionic species into PPy films to ensure stable response. A premade ionic polymer is not optimal in many cases, as it enforces its own structure on the conducting polymer, therefore, polymerization during fabrication is preferred. Pyrrole (Py) was electropolymerized at low temperature together with a polymerizable ionic liquid (PIL) monomer in a one-step polymerization, to form a stable film on the working electrode. The structure and morphology of the PPyPIL films were investigated by scanning electron microscopy (SEM), energy dispersive X-ray spectroscopy (EDX), Fourier-transform infrared (FTIR) spectroscopy and solid-state NMR (ssNMR) spectroscopy. The spectroscopy results confirmed the successful polymerization of Py to PPy and PIL monomer to PIL. The presence of (TFSI^–^) anions that balance the charge in PPyPIL was confirmed by EDX analysis. The electrical properties of PPyPIL in lithium bis(trifluoromethanesulfonyl)-imide (LiTFSI) aqueous and propylene carbonate solutions were examined with cyclic voltammetry (CV), chronoamperometry, and chronopotentiometry. The blend of PPyPIL had mixed electronic/ionic conductive properties that were strongly influenced by the solvent. In aqueous electrolyte, the electrical conductivity was 30 times lower and the diffusion coefficient 1.5 times higher than in the organic electrolyte. Importantly, the capacity, current density, and charge density were found to stay consistent, independent of the choice of solvent.

## 1. Introduction

π-conjugated electrically conducting polymers (CP) like polypyrrole (PPy), polyaniline (PANI), and poly(3,4-ethylenedioxythiophene) (PEDOT) have shown attractive properties that make them suitable for applications in flexible electronics [1], bio-sensors [2,3,4], capacitors [5], batteries [6], and soft actuators [7,8,9]. One of the most studied conducting polymers has been PPy, because of its ease of preparation, high electrical conductivity, biocompatibility [10], and electromechanical activity [11]. PPy is synthesized via the oxidation of pyrrole, which can be done chemically or electrochemically. The electrochemical polymerization is controlled by synthesis temperature [12,13], current density [14], electrode material [15], solvent [16,17], monomer concentration [18], and—importantly—the counter ions [19]. These parameters determine the properties of the resulting polymer, including structure, conductivity, specific (pseudo)capacitance, and ion selectivity. The latter becomes important upon cycling, as the ions from the electrolyte solution move in or out of the polymer network to compensate the charge. Whether it is an anion- or cation-active process depends on the electrolyte as well as on the solvent, as shown recently [16,20]. The solvent has a strong influence on the electrochemical and electromechanical processes in any conducting polymer-based material. One way to have more control over the anion- or cation-active reactions and lessen the solvent and electrolyte effects is to block the motion of one type of ions.

The most well-known system of polyanions and conducting polymers is the PEDOT:PSS, where polystyrene sulfonate (PSS) plays the role of the immobile anions. In the commercially available aqueous suspension form, PEDOT:PSS can be printed [21] or otherwise coated on substrate. However, even the properties of PEDOT:PSS have been shown to be extremely sensitive towards solvent and additional dopants [22], as the high conductivity arising from local phase separation of PEDOT nanocrystals is easily affected. While providing several valuable benefits, PSS is not an optimal polyanion in many respects. 

Polymerized ionic liquids are a novel type of solid electrolytes, where anions or cations are immobilized as polyanions or polycations, leaving only their counter ions mobile. Polymerized ionic liquids combine the unique conductive, thermostable and ionic properties of ionic liquids (ILs) with the mechanical properties and improved processability of polymers [23,24,25]. However, the ionic conductivity of a polymerized ionic liquid is generally lower than that of polymer–ionic liquid mixtures [26], where both cations and anions are mobile [24,25].

The combination of PIL with conducting polymers has been described before, with various synthesis and objective approaches, such as applying monomers with imidazolium cations to electropolymerize thin films [27,28]. Chemical polymerization of conducting polymers in aqueous solution using hydrophobic PIL as a stabilizer has allowed to make entrapped conducting polymers inside the microparticles [29] for potential applications in OLED [25]. Recently, vapor polymerization of a conducting polymer with PIL was presented, to make materials for supercapacitors [30]. Combinations of zwitter-ionic PILs (PZILs) with covalent linkage to polypyrrole graphene-oxide layers or on gold nanoparticles with polypyrrole nanotubes have been presented for sensor applications [31,32]. 

Therefore, the combination of polymerized ionic liquids and CPs in a single material presents an interesting approach for developing ionic electroactive materials with various beneficial characteristics.

In this study, PPy was electropolymerized in a polymerizable ionic liquid solution with the goal to obtain an embedded polyelectrolyte that can (a) embedded enhance the ion mobility, especially upon reduction of the conducting polymer; and (b) behave as an ion-selective component carrying localized positive charges which leads to anion-dominated activity of the conducting polymer. The formed polymer blend was analyzed using solid state nuclear magnetic resonance (ssNMR) spectroscopy to confirm the material composition. Cyclic voltammetry (CV) and chronopotentiometry were used to study the electrical properties and solvent effects of the novel blend in the bulk; and modified scanning ionic conductance microscopy (mSICM) [33] was applied to evaluate the properties at the film surface.

## 2. Materials and Methods

### 2.1. Materials

Propylene carbonate (PC, 99%) and ethanol (analytical standard) where purchased from Sigma-Aldrich and used as received. Pyrrole (98% Sigma-Aldrich, Taufkirchen, Germany), was distilled prior use and stored at low temperature. Lithium bis(trifluoromethanesulfonyl)imide (LiTFSI, 99.95%) was acquired from Solvionic (Toulouse, France). The polymerizable ionic liquid (1-vinyl-3-hexylimidazolium bis(trifluoromethylsulfonyl)imide) was synthesized by N-alkylation of 1-vinylimidazole with bromohexane [34] followed by anion metathesis [35]. The exact parameters and reaction conditions of the synthesis of the polymerizable ionic liquid are presented in the Appendix A (Appendix A).

### 2.2. Electrochemical Synthesis of PPyPIL

Electrochemical synthesis of PPyPIL (polypyrrole-polymerized ionic liquid) was carried out galvanostatically (controlled with a PARSTAT 2273 potentiostat FRA, Princeton Applied Research, Berwyn, PA, USA) in a two-electrode electrochemical cell at 0.1 mA cm^−2^ for 40,000 s at −20 °C. The polymerization solution of PPyPIL consisted of 0.1 M Py and 0.1 M PIL monomer solution in PC. The galvanostatic polymerization of PPyPIL was carried out using a 30 × 30 mm stainless steel (AISI 316L) plate as the anode and a stainless-steel mesh (AISI 316L) as the cathode. After polymerization, the polymer film was removed from the anode and washed with ethanol and distilled water to remove the residues of Py and PIL monomer from the surface. The free-standing films were dried in a vacuum oven and stored between two glass plates in thickness of 38 µm.

### 2.3. Characterization

Fourier-transform infrared (FTIR) and solid-state nuclear magnetic resonance (ssNMR) spectroscopy were used to obtain structural information about the samples. The FTIR measurements were performed using a Bruker Alpha spectrometer using a Platinum ATR. The films were analyzed over the range of 3500–500 cm^−1^. ^13^C MAS-NMR spectra were recorded on a Bruker AVANCE-II spectrometer with a 14.1 T magnet (^13^C resonance at 150.47 MHz) using an in-house MAS_NMR double resonance probe for 4 mm rotors. The samples where powdered, mixed with silica gel L600 insulating powder and rotated at 12.5 kHz. For PPyPIL, the standard cross polarization pulse sequence was used with 0.5 ms cross polarization pulse and subsequent proton decoupling. The ion contents were verified with energy-dispersive X-ray spectroscopy (EDX) (Oxford Instruments with X-Max 50 mm^2^ detector). Fresh unactuated films where broken in liquid nitrogen to study the cross-sections of the samples. The surface conductivity of the films was determined by four-point probe method using a surface resistivity meter (Guardian Manufacturing Inc., Cocoa, FL, USA).

For studying the electrochemical properties of PPyPIL, rectangular samples of 1.5 × 0.1 cm were cut from the synthesized films. Cyclic voltammetry (5 mV s^−1^) and chronopotentiometry of the PPyPIL samples were performed in the potential range of 0.65 to −0.2 V. The charge obtained from current density-time curves was used for calculating the diffusion coefficient as given by Equations (1) and (2) [36].
(1)$$ln\left[1-\frac{Q}{Q_{t}}\right]=-bt$$
where Q is the charge consumed at time t, and $Q_{t}$ is the total charge consumed, calculated from the integration of the current-time curve (from the chronoamperometry experiments). The diffusion coefficient D is included in the constant b, as given by Equation (2), where h is the thickness of PPyPIL films (38 μm):(2)$$D=\frac{bh^{2}}{2}$$

Plotting $ln\left[1-\frac{Q}{Q_{t}}\right]$ versus t gives the constant b as the slope [36], and if the film thickness is known, the diffusion coefficient *D* can be calculated. Specific capacitance is obtained from the chronopotentiometry measurements using Equation (3):(3)$$C_{S}=\frac{i}{-slope\cdot m},$$
where i is the discharging current and m the mass of the PPyPIL sample. The slope (ΔV/Δt) was calculated after the IR drop from the potential time curves at discharging. The mass of the PPyPIL films in LiTFSI-PC and LiTFSI-aq was 535.8 ± 43 µg and 401.9 ± 32 µg, respectively.

Modified scanning ionic conductance microscopy (mSICM) [33] was used to study the electrochemical properties on the surface of PPyPIL. mSICM cyclic voltammetry measurements (0.65 V to −0.2 V) were carried out in LiTFSI-aq and LiTFSI-PC with 50 mV s^−1^ scan rate. mSCIM uses single-barreled micropipettes with tip diameters of 5–10 µm, fabricated from 100 mm long, 2 mm outside diameter borosilicate glass capillaries (Harvard Apparatus product number 30-0117) using a Sutter Instruments P-2000 laser puller. The micropipette containing the electrolyte and a platinum wire as the counter electrode (CE) was moved over PPyPIL film surface. The polymer was connected as the working electrode (WE) accompanied by a reference electrode (Ag/AgCl wire, RE). The electrolyte from the pipette forms a droplet on PPyPIL surface where ionic fluxes can be measured (Appendix A).

## 3. Results and Discussion

### 3.1. PPyPIL Films

The synthesis chronopotentiogram is shown in Figure 1a and the SEM images of the surface and the cross-section of the films are shown in Figure 1b. The chronopotentiogram of PIL electropolymerization are shown in Appendix A. The EDX spectrum of the PPyPIL film is shown in Figure 1c.

The potential-time curves (Figure 1a) of PPyPIL galvanostatic polymerization were smooth and uniform. During the first 2 h, the polymerization potential reached the maximum of 3.34 V. Compared to the electrochemical synthesis of pure polypyrrole under the same conditions, the potential was significantly higher, which we assume relies on the limited ionic conductivity of the PIL, found in range of 10^−6^ S cm^−1^ [25], which is about 1000 times lower than the usual ionic liquids in range of 10^−3^ S cm^−1^ [37]. The potential then started to slightly decrease, to reach 3.28 V by the end of the polymerization process. This behavior indicates a rather uniform PPyPIL deposition [38]. The surface morphology of the synthesized film under SEM (Figure 1b) was rough and globular which is common for polypyrrole synthesized in ionic liquids [39]. The structure at the cross-section of PPyPIL (inset of Figure 1b), however, was dense and homogeneous, which is not usual for PPy free standing films.

The EDX spectrum (Figure 1c) displays the peaks of carbon (C) at 0.26 keV, nitrogen (N) at 0.38 keV, oxygen (O) at 0.52 keV, fluorine (F) at 0.68 keV, and sulfur (S) at 2.32 keV. The oxygen, fluoride and sulfur peak were assigned to the TFSI^–^ anions compensating the positive charges of the PPyPIL. The surface conductivity of the PPyPIL films immersed in LiTFSI-aq was 2.0 ± 0.1 S cm^−1^ and in LiTFSI-PC 2.6 ± 0.2 S cm^−1^, clearly lower than the 61 ± 5 S cm^−1^ for pure PPy/TFSI films [40].

### 3.2. Structural Analysis of PPyPIL with ssNMR and FTIR

The electropolymerization of Py in the presence of PIL monomers formed a stable PPyPIL film on the working electrode. To determine the composition of the PPyPIL films, ssNMR spectroscopy was performed, the spectra are shown in Figure 2a and the FTIR spectra presented in Figure 2b. 

Figure 2a shows the comparison between the NMR spectra of the PIL-monomer and PPyPIL after electropolymerization. The typical signals of the vinyl group of the PIL monomer were found at 109.6 ppm (108 ppm in literature [41]), but not observed for PPyPIL revealing that the polymerization of the PIL-monomer indeed took place during the electrodeposition. From the imidazolium ring, the C1 atom can be found at 135 ppm (137 ppm in literature [42]) and the C2 and C3 of imidazolium are represented by several peaks between 120–124 ppm. The peaks at 38–54 ppm relate to the polymer chain (C4 and C5 in PPyPIL) and signals around 14–33 ppm, at 14.2 and 23.1 also found in the PPyPIL spectrum, was assigned to the carbon atoms (C6–C11) of the 3-hexyl substituent of imidazolium [41]. The main PPy resonance peak at 126–128 ppm, corresponding to predominant α—α’ bonding [43] is represented here at 127 ppm in case of PPyPIL. The quinoid structure of oxidized PPy [44] is represented by a peak at 142.3 ppm for PPyPIL. The 150.3 ppm peak was assigned to the C1 atom of the imine group [45]. From the solid state ^13^C MAS NMR results, it can be concluded that signals from both PPy groups and PIL groups were present in PPyPIL. The connection of pyrrole over the amino group with PIL units (structure B, Appendix A) was unlikely, as Martina et al. have shown [46] that the pyrrole units with N substitutes become twisted by 70°, and therefore, the conjugation system is interrupted. Connection in the β position of the pyrrole units (structure C, Appendix A) was not detected either, as the characteristic peaks at 106 ppm and 114 ppm [44] were not found in the PPyPIL spectrum.

Figure 2b shows the FTIR spectra of the PIL monomer and PPy-PIL. The main questions were if the polymerization of PIL monomer took place, answered by the disappearance of the vinyl group signals in the PPy-PIL spectrum. The 3072 cm^−1^ peak (inset in Figure 2b) shows the (=C–H stretching [47]) vinyl group, the 914 cm^−1^ out of plane =C–H bending [47] and the 786 and 760 cm^−1^ peaks represent the H-C=C–H wagging [47], which have disappeared after the electropolymerization. Therefore, the polymerization of PIL monomer together with pyrrole took place. The C–H stretching from imidazole ring [48] of PPy monomers are found between 2800 and 3200 cm^−1^ also seen as small waves for PPy-PIL. Additional peaks of the imidazole ring can be observed from C–N stretching vibration between 1550–1660 cm^−1^ with the 1660 cm^−1^ peak shown as well in PPy-PIL spectrum. The counter anion TFSI^–^ has been detected previously [49] at 1228 cm^−1^ (shoulder), 1172 cm^−1^ and 1052 cm^−1^. 

The peaks at 1515 cm^−1^ and 1425 cm^−1^ were attributed to the PPy ring vibrations and the band at 1283 cm^−1^ to the CH in plane vibrations [50]. The C=C stretching of aromatic compounds appears between 1000–1100 cm^−1^, observed at PPy-PIL at 1007 cm^−1^. The 1120 cm^−1^ is assigned to S=O groups [51] which we assume shows the content of TFSI^–^ as the immobilized species in PPy-PIL networks.

Considering the possible structures shown in Appendix A, the PIL contains TFSI^–^ anions to balance the localized positive charges of the imidazolium derivate, together with the anions compensating the charge of the oxidized PPy. Considering FTIR, ssNMR and EDX spectra, we propose a structure of the PPyPIL shown in Figure 3.

The covalent connection between PPy and PIL polymers (compound D and C in Appendix A) is unlikely, instead, we propose that the PPyPIL is formed as a blend of two separate polymers (Appendix A, structures A and B, Figure 3). The weakly solvating TFSI^−^ counterions [52] can be, thus, shared between the PPy and PIL polymer chains. It is well known that during electropolymerization, the PPy chains are formed as partly oxidized with counterions balancing the positive charge [38]. The charges on the PIL polymer chains are localized, while those on the PPy are delocalized.

### 3.3. Modified SICM Measurements of PPyPIL Films

Modified scanning ionic conductance microscopy was used for the electrochemical characterization of the synthesized PPyPIL films in LiTFSI-aq and LiTFSI-PC electrolytes, under cyclic voltammetry (Figure 4) to provide a more sensitive response and also to assess possible local inhomogeneitys.

Incidentally, the surface mSICM response was rather uniform, no domains of predominantly PPy or PIL content were identified. The shape of the CV curves measured on the surface of PPyPIL in LiTFSI-PC (Figure 4a) showed three rather small oxidation peaks: a broad wave at 0.14 V with two additional waves at 0.36 V and 0.43 V, revealing different stages of oxidation. The reduction peaks were found as even smaller waves at 0.3 V and 0.06 V. In LiTFSI-aq, only one oxidation peak can be distinguished at 0.43 V, as well as a reduction wave at −0.04 V. The electronic conductivity of the samples in LiTFSI-PC was 23% higher compared to those in aqueous electrolyte, reflected here in the somewhat higher charge per half cycle (Figure 4b): LiTFSI-PC 26 μC vs. LiTFSI-aq 20 μC. The electrochemical cycling of PPyPIL samples took place at steady state conditions evidenced by closed loops of charging/discharging curves [53]. In comparison, the calculated current densities and the charge densities of PPyPIL films in bulk (Appendix A) were indifferent to solvents. Additionally, no oxidation or reduction peaks were discovered in this voltage range with bulk PPyPIL films showing nearly capacitive behavior (Appendix A).

### 3.4. Potential (Current) Square Wave Step Measurements of PPyPIL Films

Square wave potential step measurements of PPyPIL films in LiTFSI-PC and LiTFSI-aq electrolytes were performed at 0.65 V to −0.2 V. The charge densities are shown in Figure 5a and the diffusion coefficient at oxidation in Figure 5b.

The current density curves of PPyPIL films in LiTFSI-PC and LiTFSI-aq at 0.0025 Hz (Appendix A) were nearly identical. This was also reflected in charge density (Figure 5a) upon reduction and oxidation at frequencies 0.0025–0.1 Hz: the charge densities were practically the same in LiTFSI-aq and LiTFSI-PC. The highest charge density in the applied frequency range upon oxidation was observed at 0.0025 Hz: 42 C cm^−3^ in the case of LiTFSI-PC (40 C cm^−3^ for LiTFSI-aq). It has been observed before that the charge densities at oxidation/reduction of PPy [54] or PEDOT films [55] tend to be higher in aqueous electrolytes than in propylene carbonate. It has been explained by higher ionic conductivity of the aqueous electrolyte compared to propylene carbonate. In this study, however, the solvent had no significant effect in view of charging/discharging behavior, likely related to the counter-ion transport along the PIL chains, without significant solvation.

Figure 5b shows that the diffusion coefficient of PPyPIL on oxidation was still dependent on the solvent. In LiTFSI-aq, the diffusion coefficient D_ox_ at 0.1 Hz was 1.5 times larger than that in LiTFSI-PC. A similar tendency could be seen during the reduction of PPyPIL (Appendix A) with the diffusion coefficients D_red_ higher in LiTFSI-aq than in LiTFSI-PC, partially explained by the larger solvent uptake in PC.

To calculate the specific capacitance C_S_ (presented in Figure 6b) of the PPyPIL films (Equation (3)) in LiTFSI-aq and LiTFSI-PC, the current square wave measurements at frequency range 0.0025 Hz–0.1 Hz were performed with the applied currents between 0.05 mA to 2 mA (same charge of 10 mC). The potential time curves of 2nd and 3rd cycle at frequency 0.0025 Hz (0.05 mA) are shown in Figure 6a.

Figure 6a shows that the square wave current steps of the 2nd and 3rd cycles of PPyPIL films gave identical response in both electrolytes, which confirms that charging/discharging was in balance, as seen before [20,53]. The results of the specific capacitance of the films showed that despite the different solvents, the values were close—in the range of 75 ± 2.4 F g^−1^ at 0.0025 Hz and, as expected, decreasing with increasing frequency [56]. Conducting polymers are defined as pseudo-capacitors due to their redox reactive nature [57], while strongly dependent on film thickness, specific capacity values of 100 F g^−1^ have been reached with PPy in LiTFSI-aq electrolyte under similar conditions [58]. As the PIL part of the material is not participating in the redox charging and with thicker films, the specific capacitance cannot reach as high, but on the other hand, the positively charged polycations inside the PPyPIL make the electrochemical behavior much more stable and also independent of the solvent.

## 4. Conclusions

In an attempt to create a more stable and consistently performing electroactive material, novel polypyrrole–polymerized ionic liquid freestanding films were successfully obtained in one-step electropolymerization. The ssNMR and FTIR spectra confirmed the successful concurrent polymerization of PIL together with PPy, with a proposed blend structure. SEM images revealed a rough and globular surface modification and EDX spectroscopy identified TFSI^–^ anions inclusion to balance the positive charges of PPyPIL upon oxidation. In the organic electrolyte solution, the electrical conductivity was 30 times higher, while the diffusion coefficients were 1.5 times higher in the aqueous electrolyte solution.

Several solvent and electrolyte effects, usually manifested in conventional electropolymerized PPy but often even PEDOT:PSS, like changes in capacity, current density and charge density, were significantly subdued in the novel PPyPIL. The choice of solvent played almost no role, the values of most electrochemical parameters were virtually the same in both aqueous and organic solutions despite significantly different solvent uptake.

This work provides a reliable method to produce new multifunctional blend materials: by concurrent polymerization, the combination of the electroactive properties of polypyrrole and the unique properties of polymerized ionic liquids are combined. Such materials can be applied in the development of flexible electronics, bio-sensors, capacitors, batteries, and actuators adding an extra dimension of control over stability and other properties of the electroactive materials.

## Figures and Tables

**Figure 1 polymers-12-00136-f001:**
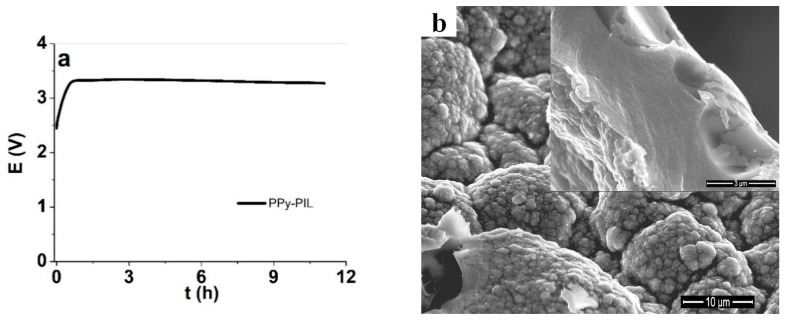
(**a**) Chronopotentiogram of PPyPIL galvanostatic synthesis, (**b**) SEM image of PPyPIL surface (scale bar 10 μm) with cross-section as an inset (scale bar 3 μm) and (**c**) EDX spectrum of PPyPIL film directly after polymerization (oxidized state). PPy, polypyrrole; PIL, polymerizable ionic liquid; EDX, energy dispersive X-ray spectroscopy.

**Figure 2 polymers-12-00136-f002:**
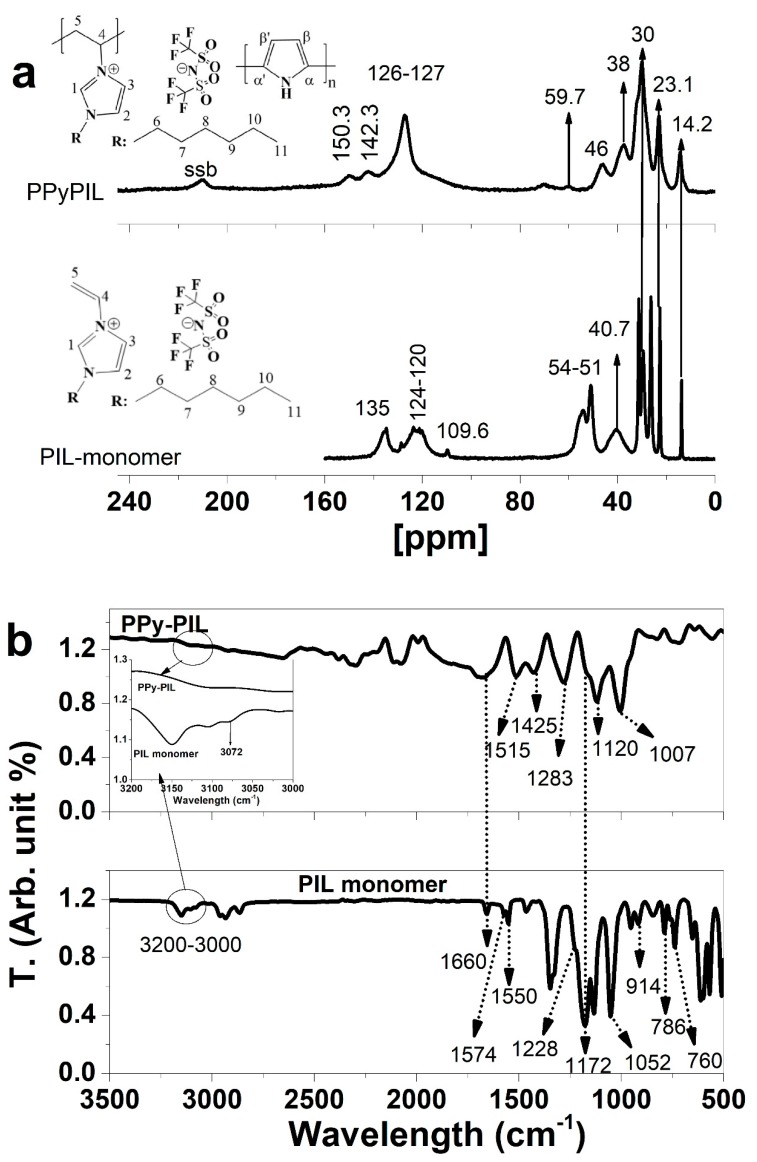
^13^C MAS-NMR spectra of electropolymerized PPyPIL with spinning side bands (ssb) and the PIL monomer are presented in (**a**). FTIR spectra of PPyPIL and PIL monomer (3500–500 cm^−1^) with inset of the 3200–3000 cm^−1^ are shown in (**b**).

**Figure 3 polymers-12-00136-f003:**
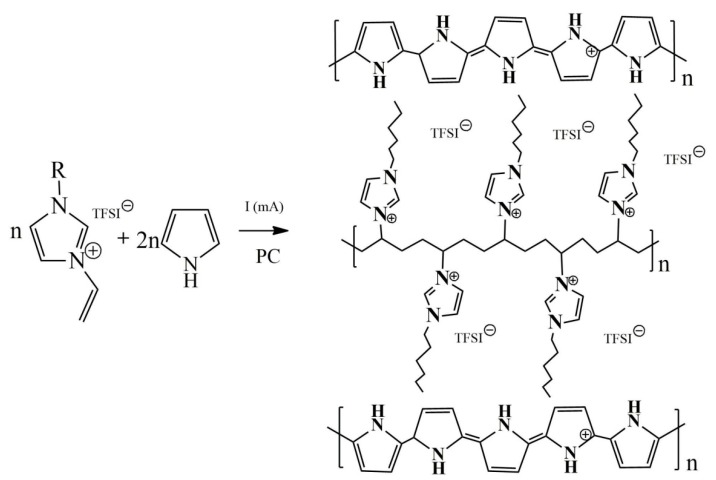
Structure of the proposed PPyPIL blends formed during the electropolymerization of pyrrole and PIL monomer in PC.

**Figure 4 polymers-12-00136-f004:**
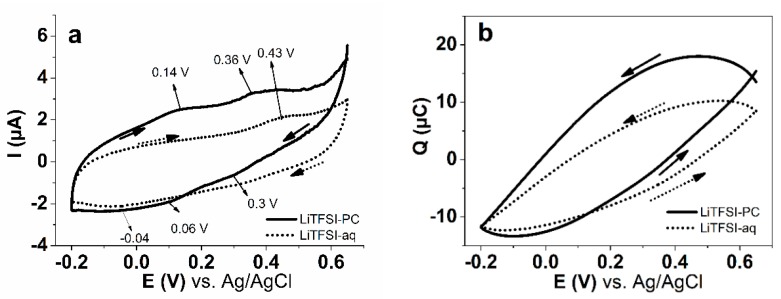
Cyclic voltammetry (50 mV s^−1^, 3rd cycle) with mSICM measurements of PPyPIL film surface in LiTFSI-aq (dotted) and in LiTFSI-PC (solid line) of (**a**) current I and (**b**) charge Q vs. potential E.

**Figure 5 polymers-12-00136-f005:**
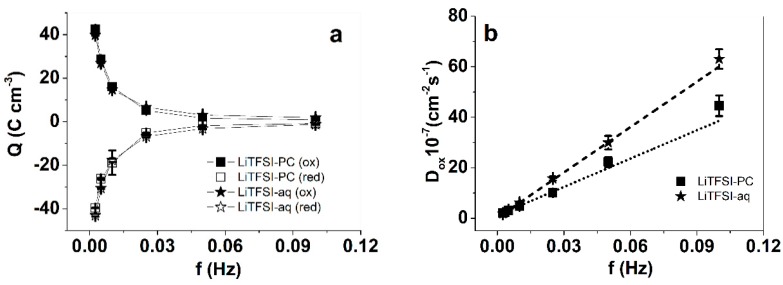
PPyPIL bulk films in potential step measurements at applied frequencies 0.0025 Hz to 0.1 Hz operating at potential 0.65 V to −0.2 V in LiTFSI-PC (■ (ox), □ (red)) and LiTFSI-aq (★ (ox), ☆ (red)) electrolytes. The charge densities Q against applied frequency f are shown in (**a**) and the diffusion coefficients at oxidation D_ox_ applying Equations (1) and (2) are shown in (**b**).

**Figure 6 polymers-12-00136-f006:**
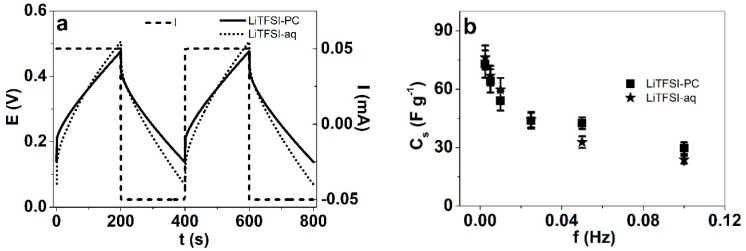
(**a**) Chronopotentiogram of PPyPIL films at 0.0025 Hz (±0.05 mA, 2nd and 3rd cycles, dashed) in LiTFSI-PC (solid line) and LiTFSI-aq (dotted) at constant charge of 10 mC. (**b**) The specific capacitance C_s_ was calculated using Equation (3) of the PPyPIL films in LiTFSI-PC (■) and LiTFSI-aq (★) against the applied frequency.

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
