# Peer review of "Improving the Electrochemical Performance and Stability of Polypyrrole by Polymerizing Ionic Liquids"

_polymers, 2020, doi:10.3390/polym12010136_

Round 1

Reviewer 1 Report

Introduction section should be improved by focusing on the purposes of the study. Accordingly, more relevant articles especially recent studies should be referred in the section to support electrochemical performance and stability of PPy. Whenever working with coupled materials, I suggest comparing the efficiency between the pure material and the coupled. Perhaps the pure material is as effective as the coupled. So, I suggest modified SICM measurements of PPy in both aqueous and organic solvents. Only scientific findings, and discussions or conclusions of them can be cited. So, the references in page 6, last paragraph are not acceptable. The conclusion need re-written. It does not contain all the relevant facts.

Author Response

We thank the reviewer for the positive review and have modified the manuscript following his/her suggestions

Introduction section should be improved by focusing on the purposes of the study.

Although we have discussed the benefits of embedded ionic and ion-selective polymeric additives, we have included a more precise goal definition in the introduction

In this study, PPy was electropolymerized in a polymerizable ionic liquid solution with the goal to obtain an embedded polyelectrolyte that a) embedded enhance the ion mobility, especially upon reduction of the conducting polymer; and b) behave as an ion-selective component carrying localized positive charges which lead to anion-dominated activity of the conducting polymer.

Accordingly, more relevant articles especially recent studies should be referred in the section to support electrochemical performance and stability of PPy.

We have included in introduction new references 10 and 11

Ghanbari, R.; Ghorbani, S.R.; Arabi, H.; Foroughi, J. The charge transport mechanisms in conducting polymer polypyrrole films and fibers. Mater. Res. Express 2018, 5, 1–6.

Bober, P.; Capáková, Z.; Acharya, U.; Zasońska, B.A.; Humpolíček, P.; Hodan, J.; Hromádková, J.; Stejskal, J. Highly conducting and biocompatible polypyrrole/poly(vinyl alcohol) cryogels. Synth. Met. 2019, 252, 122–126.

Whenever working with coupled materials, I suggest comparing the efficiency between the pure material and the coupled. Perhaps the pure material is as effective as the coupled.

We thank the reviewer for this excellent point. Having PPy electropolymerized, the solvent and the electrolyte plays a crucial role of its performance, for example if using PPy polymerized in TBACF3SO3 propylene carbonate anion and cations are exchanged during reversible redox reactions. The inclusion of PIL establishes a pure anion driven PPyPIl complex independent of the solvent. The pure PPy with different electrolytes or solvent differ in a great extent. A significant number of directly comparable “pure” systems have been thoroughly studied as reported in the literature, these results have been used as comparison.

So, I suggest modified SICM measurements of PPy in both aqueous and organic solvents. Only scientific findings, and discussions or conclusions of them can be cited. So, the references in page 6, last paragraph are not acceptable.

We have included the introduction of mSICM into the supplementary, however, it is greatly surprising that for some citing methods or approaches appears incorrect. There are surely thousands of publications where particular laws, formulae, methods or experimental setups have been used without explicit definition or description, instead a citation to the original use, derivation, or sth alike is provided and such approaches have been found adequate enough. There is a balance between overwhelming the reader with details and providing adequate information.

The conclusion need re-written. It does not contain all the relevant facts.

Sadly, not enough information was provided here as to which missing facts are being referred to. No conclusion can ever present all results, however, we have attempted to improve the conclusion as follows:

The ssNMR and FTIR spectra confirmed the successful concurrent polymerization of PIL together with PPy, with a proposed blend structure. SEM images revealed a rough and globular surface modification and EDX spectroscopy identified TFSI- anions inclusion to balance the positive charges of PPyPIL upon oxidation. In the organic electrolyte solution, the electrical conductivity was 30 times higher, while the diffusion coefficients were 1.5 times higher in the aqueous electrolyte solution.

Reviewer 2 Report

The authors present an interesting study on the development of polypyrrole based electroactive materials using a polymerizable ionic liquid and an electropolymerization reaction. PPyPILs are promising for applications such as flexible electronics, bio-sensors, capacitors, batteries and actuators and as such, their improvement requires further studies.

This is an interesting work, and fits within the scope of POLYMERS, however I recommend some revisions before the manuscript polymers-669351can be accepted for publication.

1) The authors claim that the combination of polymerized ionic liquids and CPs in a single material presents an interesting approach for developing ionic electroactive materials. Indeed, this is true, however there are already some reports in literature that follow this trend, for instance, polymeric ionic liquids as stabilizers in conducting polymers (see Macromol. Rapid Commun., 2005, 26: 1122-1126); poly(ionic liquids) and functionalized polypyrrole nanotubes (see Materials Science and Engineering: C, 2016, 65, 143 and Materials Science and Engineering: C, 2017, 75, 495). These studies are only some examples that obtain similar materials through different synthetic procedures. Accordingly, a more thorough introduction should mention existing reports on PPy/PILs.

2) In the electropolymerization reaction, equimolar solutions of pyrrole and IL monomer are added and polymerized. The authors characterize the respective PPyPIL using SEM, EDX and ssNMR, however more details are still needed to better understand this material. Is the ratio PPy/PIL maintained in the final product? Both polymerizations proceed at the same rate? In the end residues of Py and PIL monomer are washed off, how can we be certain that the ratio is the same?

3) In the ssNMR analysis, the PPyPIL structure should also be numbered for the assignment to be clearer. Spinning side bands (ssb) appear in PPyPIL spectrum, should be indicated as such in the figure caption. ssNMR lines for polymers are broaden due to their relaxation, for this reason the absence of IL monomer in the final polymer, the 109.6 ppm chemical shift cannot be unequivocally stated. Alternative characterization should be attempted (ATR-IR for example).

4) In the ssNMR analysis the authors state “In the PPyPIL spectra, a peak at 68.9 ppm was observed, assigned to the carbons of the dopant (TFSI- anions), seen previously at 68.3” [36]. This reference is related to polypyrrole only, no TFSI appears. Do the authors mean the 13C chemical shift for this anion? Usually the CF3 groups appear to much lower field, not at 68 ppm.

5) No doubt is interesting that the capacity, current density and charge density were found to stay consistent, independent of the choice of solvent, the electrical properties of PPyPIL, should be analysed in comparison to similar materials, to better understand the effect of the synthetic procedure in the production of the present ones.

Author Response

We thank the reviewer for his positive evaluation and have revised the manuscript according to the suggestions.

The authors present an interesting study on the development of polypyrrole based electroactive materials using a polymerizable ionic liquid and an electropolymerization reaction. PPyPILs are promising for applications such as flexible electronics, bio-sensors, capacitors, batteries and actuators and as such, their improvement requires further studies.

This is an interesting work, and fits within the scope of POLYMERS, however I recommend some revisions before the manuscript polymers-669351, can be accepted for publication.

1) The authors claim that the combination of polymerized ionic liquids and CPs in a single material presents an interesting approach for developing ionic electroactive materials. Indeed, this is true, however there are already some reports in literature that follow this trend, for instance, polymeric ionic liquids as stabilizers in conducting polymers (see Macromol. Rapid Commun., 2005, 26: 1122-1126); poly(ionic liquids) and functionalized polypyrrole nanotubes (see Materials Science and Engineering: C, 2016, 65, 143 and Materials Science and Engineering: C, 2017, 75, 495). These studies are only some examples that obtain similar materials through different synthetic procedures. Accordingly, a more thorough introduction should mention existing reports on PPy/PILs.

We thank the reviewer for his suggestion and have included new references 27 - 32 in the introduction of PPy/PIL.

The combination of PIL with conducting polymers has been described before, with various synthesis and objective approaches, such as applying monomers with imidazolium cations to electropolymerize thin films [27,28]. Chemical polymerization of conducting polymers in aqueous solution using hydrophobic PIL as a stabilizer has allowed to make entrapped conducting polymers inside the microparticles [29] for potential applications in OLED [25]. Recently, vapor polymerization of a conducting polymer with PIL was presented, to make materials for supercapacitors [30]. Combinations of zwitter-ionic PILs (PZILs) with covalent linkage to polypyrrole graphene-oxide layers or on gold nanoparticles with polypyrrole nanotubes have been presented for sensor applications [31,32]. 

Therefore, the combination of polymerized ionic liquids and CPs in a single material presents an interesting approach for developing ionic electroactive materials with various beneficial characteristics.

2) In the electropolymerization reaction, equimolar solutions of pyrrole and IL monomer are added and polymerized. The authors characterize the respective PPyPIL using SEM, EDX and ssNMR, however more details are still needed to better understand this material. Is the ratio PPy/PIL maintained in the final product? Both polymerizations proceed at the same rate? In the end residues of Py and PIL monomer are washed off, how can we be certain that the ratio is the same?

We thank the reviewer for this excellent question. No claim was made in the manuscript that the actual initial ratio is maintained in the polymerized material. As a matter of fact, to determine the ratio is rather difficult and it probably can be influenced with a variety of synthesis conditions. To establish a clear ratio and ways to influence such would make a nice research objective in itself, but was unfortunately out of scope for the present study. ssNMR and FTIR results indicated that both were included in the blend structure but we have not quantified the ratio. We have added the polymerization curve of pure PIL in the supplementary (Figure S4), but due to different standard potentials and the ionic conductivity of the monomer solution, we assume that polymerization does not proceed at the same rate during copolymerization.

3) In the ssNMR analysis, the PPyPIL structure should also be numbered for the assignment to be clearer.

Numbers have been added.

Spinning side bands (ssb) appear in PPyPIL spectrum, should be indicated as such in the figure caption.

The term has been added in the Figure capture

 ssNMR lines for polymers are broaden due to their relaxation, for this reason the absence of IL monomer in the final polymer, the 109.6 ppm chemical shift cannot be unequivocally stated. Alternative characterization should be attempted (ATR-IR for example).

Indeed, broader peaks are clearly apparent. We agree with the reviewer and have performed FTIR measurements shown in Figure 2b

Figure 2b shows the FTIR spectra of the PIL monomer and PPy-PIL. The main questions were if the polymerization of PIL monomer took place, so the vinyl group signals disappear in PPy-PIL spectrum. The 3072 cm-1 peak (inset in Figure 3b) shows the (=C-H stretching [47]) vinyl group, the 914 cm-1 out of plane =C-H bending [47] and the 786 and 760 cm-1 peaks represent the H-C=C-H wagging [47], which have disappeared after the electropolymerization. Therefore, the polymerization of PIL monomer together with pyrrole took place. The C-H stretching from imidazole ring [48] of PPy monomer are found between 2800 and 3200 cm-1 also seen in small waves for PPy-PIL. Additional peaks of the imidazole ring can be observed from C-N stretching vibration between 1550 -1660 cm-1 with the 1660 cm-1 peak shown as well in PPy-PIL spectrum. The counter anion TFSI- has been detected previously [49] at 1228 cm-1 (shoulder), 1172 cm-1 and 1052 cm-1.

The peaks at 1515 cm-1 and 1425 cm-1 were attributed to the PPy ring vibrations and the band at 1283 cm-1 to the CH in plane vibrations [50]. The C=C stretching of aromatic compounds appears between 1000-1100 cm-1, observed at PPy-PIL at 1007 cm-1. The 1120 cm-1 is assigned to S=O groups [51] which we assume shows the content of TFSI- as the immobilized species in PPy-PIL networks.

4) In the ssNMR analysis the authors state “In the PPyPIL spectra, a peak at 68.9 ppm was observed, assigned to the carbons of the dopant (TFSI- anions), seen previously at 68.3” [36]. This reference is related to polypyrrole only, no TFSI appears. Do the authors mean the 13C chemical shift for this anion? Usually the CF3 groups appear to much lower field, not at 68 ppm.

We thank the reviewer for pointing this out, the incorrect interpretation as well the wrong citation were removed.

5) No doubt is interesting that the capacity, current density and charge density were found to stay consistent, independent of the choice of solvent, the electrical properties of PPyPIL, should be analysed in comparison to similar materials, to better understand the effect of the synthetic procedure in the production of the present ones.

We thank the reviewer for this valuable comment. Indeed, it would be great if direct comparisons could be made with similar materials. Unfortunately, accurate numeric comparisons would require to first establish comparable (and balanced) electrochemical windows for all systems, as both material properties (conductivity, thickness, density) as well as approach (electrochemical window, scanrate, etc) can play a role. Hence, it is unlikely that a study of reference system could be found in the literature and the respective materials need to be made/obtained, characterized and studied together.  Hopefully such results can be provided in the future.

Reviewer 3 Report

In their submission to Polymers entitled “Improving the electrochemical performance and stability of polypyrrole by polymerizing ionic liquids”, Keifer and co-workers describe the fabrication of a pyrrole-based polymer by means of electropolymerization at low temperature. This polymer was prepared in combination with a polymerizable ionic liquid monomer in a single-step polymerization, yielding a film on the working electrode. The structure and morphology of the films were evaluated by scanning electron microscopy (SEM), energy dispersive X-ray spectroscopy (EDX) and solid state NMR (ssNMR). However, FT-IR would be also a characterization technique that should be used. Therefore, I recommend the authors to perform FT-IR analysis of the polymeric films. Regarding the polymerization solution, have the athours assayed other concentrations rather than those described in page 2 (0.1 M Py and 0.1 M PIL)? Th The electrical properties of PPyPIL extracted the different electrochemical techniques are very interesting but a proper comparison with previously reported systems is required. After all these considerations have been taken into account in the manuscript, I recommend its publication in Polymers.

Author Response

In their submission to Polymers entitled “Improving the electrochemical performance and stability of polypyrrole by polymerizing ionic liquids”, Keifer and co-workers describe the fabrication of a pyrrole-based polymer by means of electropolymerization at low temperature. This polymer was prepared in combination with a polymerizable ionic liquid monomer in a single-step polymerization, yielding a film on the working electrode. The structure and morphology of the films were evaluated by scanning electron microscopy (SEM), energy dispersive X-ray spectroscopy (EDX) and solid state NMR (ssNMR). However, FT-IR would be also a characterization technique that should be used. Therefore, I recommend the authors to perform FT-IR analysis of the polymeric films.

We thanks the reviewer for his positive evaluation of our manuscript and have made the revision according to the suggestions.

ATR-FTIR was performed and the results appear as Figure 2b

Figure 2b shows the FTIR spectra of the PIL monomer and PPy-PIL. The main questions were if the polymerization of PIL monomer took place, so the vinyl group signals disappear in PPy-PIL spectrum. The 3072 cm-1 peak (inset in Figure 3b) shows the (=C-H stretching [47]) vinyl group, the 914 cm-1 out of plane =C-H bending [47] and the 786 and 760 cm-1 peaks represent the H-C=C-H wagging [47], which have disappeared after the electropolymerization. Therefore, the polymerization of PIL monomer together with pyrrole took place. The C-H stretching from imidazole ring [48] of PPy monomer are found between 2800 and 3200 cm-1 also seen in small waves for PPy-PIL. Additional peaks of the imidazole ring can be observed from C-N stretching vibration between 1550 -1660 cm-1 with the 1660 cm-1 peak shown as well in PPy-PIL spectrum. The counter anion TFSI- has been detected previously [49] at 1228 cm-1 (shoulder), 1172 cm-1 and 1052 cm-1.

The peaks at 1515 cm-1 and 1425 cm-1 were attributed to the PPy ring vibrations and the band at 1283 cm-1 to the CH in plane vibrations [50]. The C=C stretching of aromatic compounds appears between 1000-1100 cm-1, observed at PPy-PIL at 1007 cm-1. The 1120 cm-1 is assigned to S=O groups [51] which we assume shows the content of TFSI- as the immobilized species in PPy-PIL networks.

Regarding the polymerization solution, have the athours assayed other concentrations rather than those described in page 2 (0.1 M Py and 0.1 M PIL)?

We thank the reviewer for this excellent question. In general, the concentrations in the reactant solution are not fully independent, as there is a requirement for sufficient conductivity (electrolyte) and the sensible ratios of the monomer and the counterions should be maintained.  We cannot report other ratios in pyrrole to PIL concentration but we will consider this in future work.

 Th The electrical properties of PPyPIL extracted the different electrochemical techniques are very interesting but a proper comparison with previously reported systems is required.

A good point. Indeed, it would be great if direct comparisons could be made with similar materials. Unfortunately, accurate numeric comparisons would require to first establish comparable (and balanced) electrochemical windows for all systems, as both material properties (conductivity, thickness, density) as well as approach (electrochemical window, scan rate, etc) can play a role. Hence, it is unlikely that a study of reference system could be found in the literature and the respective materials need to be made/obtained, characterized and studied together.   Hopefully such results can be provided in the future. Rather qualitative comparisons have been made and some citation were given to other CP where solvent changed and charge densities obtained:

It has been observed before that the charge densities at oxidation/reduction of PPy [54] or PEDOT films [55] tend to be higher in aqueous electrolytes than in propylene carbonate.

After all these considerations have been taken into account in the manuscript, I recommend its publication in Polymers.

Round 2

Reviewer 1 Report

The authors answered all my questions. I think it is suitable to be published. 

Reviewer 2 Report

The authors have improved their manuscript in accordance to the reviewer’s suggestions. The manuscript can now be accepted for publication.